# Understanding Socioeconomic Inequalities in Zero-Dose Children for Vaccination in Underserved Settings of Ethiopia: Decomposition Analysis Approach

**DOI:** 10.3390/ijerph21081086

**Published:** 2024-08-17

**Authors:** Gashaw Andargie Biks, Fisseha Shiferie, Dawit Abraham Tsegaye, Wondwossen Asefa, Frank DelPizzo, Samson Gebremedhin

**Affiliations:** 1Project HOPE, Ethiopia Country Office, Addis Ababa P.O. Box 45, Ethiopia; fshiferie@projecthope.org (F.S.); dtsegaye@projecthope.org (D.A.T.); 2Project HOPE Headquarter, 1220 19th St NW #800, Washington, DC 20036, USA; wasefa@projecthope.org; 3Bill & Melinda Gates Foundation, 500 5th Ave N, Seattle, WA 98109, USA; frank.delpizzo@gatesfoundation.org; 4School of Public Health, Addis Ababa University, Addis Ababa P.O. Box 1176, Ethiopia; sgebremedhin@projecthope.org

**Keywords:** zero-dose, immunization, Ethiopia, concentration index, decomposition analysis

## Abstract

Despite considerable global efforts to enhance vaccine distribution in low-income countries, a significant number of children remain unvaccinated, particularly in Ethiopia. The underlying socioeconomic challenges in these regions are recognized as primary contributors to the low vaccination rates. However, the reasons for this persistent disparity in Ethiopia’s remote and underserved regions need further analysis. The study employed a cross-sectional design and was conducted as part of the Project HOPE Zero-Dose Evaluation from 1 February to 31 July 2022. Concentration indices were utilized to quantify the extent of inequality, with further decomposition aimed at identifying contributing factors to this disparity. The findings underscored that populations with lower socioeconomic status encounter high numbers of children receiving no vaccinations. Key factors influencing the number of zero-dose children included distance from healthcare facilities (61.03%), economic status of the household (38.93%), absence of skilled birth assistance (20.36%), underutilization of antenatal care services (<four visits; 8.66%), lack of postnatal care (8.62%), and rural residency (7.69%). To reduce the number of zero-dose children in Ethiopia, it is essential to implement context-specific strategies that address socioeconomic barriers and integrate innovative approaches with community engagement. This approach will help to ensure equitable access to vaccines for children across all socioeconomic statuses.

## 1. Background

In developing countries, socioeconomic inequalities continue to impose barriers to achieving universal health coverage for children [1,2]. This pressing issue has captured the attention of global health policymakers, who now consider the distribution of health inequality as a paramount concern. Thus, researchers have undertaken comprehensive cross-sectional studies to shed light on the impact of socioeconomic disparities on the process of health and development.

However, we find that a dearth of evidence exists when it comes to understanding the unfolding effects of socioeconomic inequalities amidst rising urbanization, advancements in women’s education, infrastructure proliferation, and growing wealth [3,4,5], Few studies have managed to showcase the alarming truth that socioeconomic disparities in healthcare are high [6,7,8]. These disparities not only curtail the potential of underprivileged populations to actively participate in the economy but also impede their capacity to navigate life to the fullest, and, in some extreme cases, even threaten their survival.

In assessing disparities between the poor and the rich in terms of health status, a mere quantification of the disparities is insufficient [8]. It is crucial to identify the specific subgroups within the populations that are most disadvantaged [9]. To achieve this, we must unravel the factors that contribute to these inequalities, particularly those associated with people’s living conditions, such as their access to healthcare, education, and working status. 

Furthermore, a growing body of studies has taken a close look at the disparities in immunization coverage among different socioeconomic groups in developing countries like India [10,11,12]. These studies have shed light on the significance of household economic status in determining access to immunization services.

In gauging the level of socioeconomic inequalities in health, many researchers have employed concentration indices (CIs) and concentration curves. However, it is important to note that while these CIs provide an understanding of the degree of socio-economic inequality, they do not reveal the precise pathways through which such inequality manifests. Therefore, decomposition of inequalities is critical to explore pathways of socioeconomic inequalities in child health [13]. 

Over the past two decades, Ethiopia has made impressive strides in increasing childhood vaccination coverage, representing a remarkable triumph in the pursuit of better child health. Between 2000 and 2019, the proportion of children receiving all routine antigens saw a substantial rise from a mere 15% to a commendable 44%. Notably, the coverage of measles-containing vaccine 1 (MCV-1) nearly doubled, surging from 32% to an encouraging 59%. In parallel, efforts to reduce the number of children who had not received the DPT-1-containing vaccine (zero-dose) and those under-immunized (missing DPT-3-containing vaccine) yielded significant progress, with reductions of 32% and 40%, respectively. 

However, despite these achievements, crucial challenges persist. In 2019 alone, a staggering 23% of children aged 12–23 months remained zero-dose, while 39% continued to be under-immunized [14,15]. 

Alarmingly, Ethiopia ranks fourth among GAVI-supported countries in terms of the burden of zero-dose immunization [16,17,18]. This demands immediate attention and necessitates an in-depth examination to comprehend the deterring factors hindering progress in providing zero-dose children with the required immunizations. Therefore, this study explored the association between zero-dose children coverage and various socioeconomic factors and quantified their contributions to generating inequalities in zero-dose children in remote, hard-to-reach, and underserved regions of Ethiopia.

## 2. Methods

### 2.1. Study Design and Setting

We utilized the data from the Project HOPE Zero-Dose Evaluation initiative, which was funded by the Bill & Melinda Gates Foundation. The project work was undertaken in collaboration with Amref Health Africa and three local implementing partners. We undertook a quantitative cross-sectional study to address the issue of zero-dose and under-immunized children in remote, hard-to-reach, and underserved regions of Ethiopia from 1 February to 31 July 2022. Sample collection was conducted in various study settings, including pastoralist areas such as Afar, Somali, Gambella, Oromia, Southwest, and Southern Nations, Nationalities, and People’s (SNNP) regions; developing regions such as Afar, Somali, Gambella, and Benishangul Gumuz; newly formed regions like Sidama and Southwest; remote agricultural areas in Amhara, Oromia, and SNNP; conflict-impacted zones in Amhara, Afar, Oromia, and Benishangul Gumuz; urban low-income districts within major cities, including Addis Ababa, Dire Dawa, Harar, Bahir Dar, Hawassa, and Adama; and special demographic groups, specifically refugees and internally displaced persons.

### 2.2. Study Population and Sample Size

Sample sizes for each of the demographic groups were determined utilizing Cochran’s Single Proportion Sample Size Formula, considering a 95% confidence level, a margin of error of 4%, a 16% prevalence rate of zero-dose children as per CSA & The DHS Program (2017), and an additional 10% to account for potential non-responses. Consequently, it was established that a sample size of 360 was necessary for each population category. Our examination of data from DHS 2016 and Mini DHS 2019 indicated that, in Ethiopia, an average of 12 children aged 12–35 months were found per enumeration area (EA). Therefore, to achieve the required sample size of 360 children aged 12–35 months for each group, at least 30 EAs needed to be involved, presuming full participation from all children within each EA.

Table 1 outlines the initial plan to gather data from 4080 children across 340 EAs, ensuring a minimum sample size of 360 per demographic segment. However, the final survey execution faced challenges due to conflicts in several districts, resulting in a reduced participation of 3646 children aged 12–35 months from 304 EAs. Despite these hindrances, the total sample size remained sufficiently large to conduct subgroup analyses by sex and age groups.

In this study, subjects were selected through a stratified sampling method in a two-step procedure according to the recommendations of the WHO for conducting vaccination coverage cluster surveys [19]. To accurately estimate vaccination coverage, we utilized a method involving weighted analysis, which appropriately accommodates different sample sizes and accounts for linearization after stratification. The population of interest was categorized into distinct strata including urban, rural, hard-to-reach/agrarian, developing and pastoralist regions, newly established regions, internally displaced persons, and refugee camps. Each stratum was defined to be internally consistent yet distinctly different from others concerning the targeted survey metrics.

To reduce the variance in our estimates, a proportionate number of samples was assigned to each stratum, with the selection within each being conducted via simple random sampling to ensure comprehensive representation. In synthesizing our findings, we integrated results from individual strata, adjusting for potential over- or under-representation and imbalances due to stratification. Moreover, to accommodate diverse selection probabilities and differing levels of non-response, we implemented a linearized weighted analysis post-stratification. This methodology enhanced the validity and reliability of our estimates, providing a robust basis for understanding the vaccination landscape.

### 2.3. Data Collection Process

To streamline and expedite the data collection process, we used the innovative CommCare digital App [20]. This user-friendly application system, developed by Dimagi in 2022, not only met our stringent requirements but also seamlessly integrated with Project HOPE’s vast in-house capabilities and resources. One noteworthy advantage of this platform is its compatibility with major data analytics and visualization software, enabling us to collect, clean, and monitor data nearly in real time, thus ensuring the highest standards of quality and accuracy.

Data collection was performed diligently by our team of 48 experienced enumerators and 24 supervisors. To ensure the highest quality of personnel, we employed a rigorous recruitment process that considered multiple criteria. Diploma holders in health-related disciplines were given priority, along with individuals who possessed prior experience in similar surveys and proficiency in using the CommCare digital App. Successful completion of our comprehensive data collector training was also required.

Our training program consisted of a structured 5-day course delivered to enumerators and supervisors. The training covered various aspects, including an explanation of the sampling approach, fundamental principles of data collection, a line-by-line discussion on the questionnaire, hands-on practice with the CommCare digital App, mock interviews, field practice, and a review of basic ethical practices related to research involving human subjects. This comprehensive training ensured that our team was well-equipped to handle the challenges of accurate and ethical data collection.

To determine vaccination status, we employed a triangulation approach, utilizing three different sources of information: caregiver’s report, home-based vaccination cards, and facility-based reports. This multi-faceted approach enhanced the reliability and accuracy of our results, ensuring a comprehensive assessment of vaccination coverage.

Dependent variables: In our analysis, we focused on the missing Penta 1 variable, which represents the proportion of individuals not receiving the DPT-1/penta1-containing vaccine. Since this variable is binary, a normalization process is necessary to construct the confidence interval and allow for quantification within the range of −1 to 1.

Independent variables: In this research, we explored multiple predictor variables that capture the diverse aspects of a family’s socioeconomic standing. Key indicators included maternal age, marital status of the mother, mother’s level of education, mother’s occupation, family’s living location, gender of the primary child, and wealth index. Additionally, antenatal care (ANC) and postnatal care (PNC) were evaluated as significant predictors.

### 2.4. Statistical Analysis

Principal Component Analysis was utilized to condense the original dataset of 41 variables into a more manageable nine factors. These components were then aggregated to create scores that were subsequently employed in the classification of children into one of five socioeconomic quintiles, ranging from the most impoverished to the most financially well-off. In order to effectively describe the primary outcome variable, we employed frequency tables and calculated percentages. 

To assess socioeconomic disparities in children who had not received any vaccinations (referred to as zero-dose children), we estimated the CI [21]. This index is defined as two times the area between the concentration curve and the line of equality. It ranges from −1 to +1, with a value of 0 indicating equality in the distribution of vaccine uptake. More specifically, the CI measures the extent to which there is a wealth-related disparity in relation to zero-dose children. In cases where there is no socioeconomic inequality, the CI will be zero.

The CI can assume either a negative or positive value, depending on the nature of the observed wealth-related inequality. When the curve lies above the line of equality, indicating a disproportionate concentration of the health variable (in this case, high numbers of zero-dose children) among the poor, the CI will be negative. Conversely, a positive value is indicative of the curve lying below the line of equality.

To calculate the CI, we utilized the formula CI = 2/y cov (h, r), where “h” represents the healthcare outcome of interest, in this case, the vaccination status of individuals, “y” is the mean of this healthcare outcome, and “r” signifies the fractional rank of an individual within the wealth distribution.

In addition to computing the CI, we also calculated the corresponding 95% confidence intervals (CIs) to provide a measure of the uncertainty associated with our estimates.

### 2.5. Decomposing Inequality

Socioeconomic inequalities in vaccination coverage are of significant concern to policymakers. While the CI is useful for quantifying wealth-related inequalities in health service utilization, it fails to provide insights into the underlying factors contributing to these disparities. In this study, we apply an innovative approach developed by Wagstaff et al. [22] to decompose the CI of vaccination coverage and identify the individual factors driving wealth-related health inequalities. 

In our research, we employed ordinary least squares regression models to rigorously analyze the impact of various explanatory variables on a child’s vaccination status. This statistical approach allowed us to estimate a linear regression model, providing clarity on how factors such as socioeconomic status, access to healthcare, parental education, and community influences may correlate with vaccination rates among children. By leveraging the ordinary least squares methodology, we aimed to produce unbiased and efficient estimators, thereby ensuring that our findings were both reliable and robust, offering meaningful insights into effective strategies for improving vaccination coverage in diverse populations. Through this analytical framework, we could comprehensively assess the determinants that were critical in influencing a child’s likelihood of being vaccinated, contributing to the broader public health discourse on enhancing immunization programs.

We first operationalize the child’s vaccination status (v) as a function of k explanatory factors (xk) in a linear regression model: v = α + Σk βk xk + ε [2], where α and β denote parameters and ε represents the error term. To decompose the CI for zero-dose children, we employed the following formula: C = Σ K (βk ®x k/μ) Ck + GCε/μ [3]. Here, μ signifies the mean of y, ®xk denotes the mean of xk, Ck signifies the CI for xk (analogous to C), and GCε represents the generalized CI for the error term (ε). Consequently, C can be expressed as a weighted sum of CI values for each explanatory factor, where the weight for xk is the elasticity of y with respect to xk (ղk = βk ®xk/μ). The residual component captured by the last term (GCε/μ) illustrates the wealth-related inequality in health that remains unexplained by systematic variations in the regressors.

Our analyses utilized the bootstrap method with 500 replications to estimate standard errors. Adjusting for sampling design (stratification and clustering) and sampling weights, we performed data analysis using STATA (V.17, Stata Corp LLC 4905 Lakeway Drive, College Station, TX 77845-4512, USA) software packages [23].

## 3. Results

### 3.1. Sociodemographic Characteristics

Table 2 presents the sociodemographic characteristics of the study participants residing in remote, hard-to-reach, and underserved settings of Ethiopia. A total of 3646 mothers/caregivers, with children aged 12 to 35 months, participated in this study. Over half (54.0%) of the respondents fell within the age range of 25 to 34 years. An equally significant proportion (59.2%) of these mothers/caregivers had not received any formal education and a vast majority (81.4%) were from rural areas, highlighting the immense challenges faced by communities lacking access to vital healthcare services. Furthermore, most (90.8%) of the respondents were married and a significant proportion (57.6%) were unemployed at the time of data collection. 

### 3.2. CI and Decomposition

The CI analysis found a negative value (CI = −0.2250), suggesting that zero-dose children belong to the most impoverished and underprivileged communities residing in remote, hard-to-reach, and underserved regions of Ethiopia (see Table 3 for details).

### 3.3. Decomposition of Socioeconomic Inequality

Table 4 presents the decomposition analysis that was carried out based on the ordinary least square regression, which indicates the elasticity, CI, and contribution of each covariate to the overall inequality for zero-dose children. Each of these outcomes was modelled separately in the results presented in Table 3. Contributors to disparities in zero-dose children based on the results of the decomposition analysis included age of mother/caregiver, child’s gender, mother’s literacy, ANC services, PNC services, skilled birth attendance, place of residence, caregiver’s employment status, marital status, child’s age in months, household wealth, sex of the head of household, number of under-five children, and availability of a health facility in the kebele.

The overall CI value indicated that the poorest and poorer socioeconomic groups in remote, hard-to-reach, and underserved settings of Ethiopia were more disadvantaged in terms of having a greater number of zero-dose children. The decomposition results in Table 4 present CIs of the variables selected for the study together with regression coefficients and percentage contributions to the inequality in zero-dose children of the different covariates. The CI decomposition revealed that not having a health facility within the kebele (61.03%), overall wealth index (38.93%), skilled birth attendance (20.36%), PNC services (8.62%), and rural residence (7.69%) contributed a large percentage to zero-dose children’s inequalities. In addition, several factors in our study made a negative contribution to the overall CI, indicating that they contributed to a reduction in the observed wealth-related inequalities of zero-dose children coverage. These factors included child’s age in months, household head’s sex, age of the respondents, caregiver’s employment status, number of under-five children, and maternal education (Table 4). 

## 4. Discussion

In this study, we set out to investigate the extent of socioeconomic disparities among zero-dose children in remote, hard-to-reach, and underserved areas of Ethiopia. Our aim was to identify the various sociodemographic, wealth index, and maternal-related factors that contribute to the prevalence of zero-dose children in these settings.

Our research found that inequality heavily favored the higher-income groups, having lower numbers of zero-dose children within the study settings. This corroborated the results of previous studies conducted in similar developing countries [3,11,24,25]. These findings also support the argument that socioeconomic status plays a significant role in determining the vaccination coverage among children aged 12 to 35 months in Ethiopia’s underserved settings.

Multiple factors contributed to the widening gap in coverage for zero-dose children, amplifying the impact of socioeconomic inequality. One of the key drivers was the limited availability of health facilities in the kebeles, or local administrative units. This scarcity not only hinders access to vital healthcare services but also exacerbates the disparities between different income brackets.

The study examined socioeconomic inequalities among children who had not received any vaccinations and found a striking concentration of such children among the most economically disadvantaged groups. This aligns with the results of a similar study conducted in Nigeria [26]. By conducting a decomposition analysis, we were able to identify various factors that contributed to this disparity in zero-dose children in remote, hard-to-reach, and underserved areas of Ethiopia.

One significant factor was the absence of a health facility in the kebele. This lack of access to nearby healthcare facilities was found to increase the likelihood of children not receiving any vaccinations. Additionally, we discovered that living a considerable distance away from the nearest health facility played a role in this disparity. Children residing in rural areas, who often face challenges in accessing healthcare services, were more likely to be zero-dose children [26,27,28,29].

Our study also confirmed previous research findings that indicated lower immunization coverage among children in rural areas compared to urban areas. This suggests that the disparity between rural and urban locations is a recurring issue in the context of immunization coverage. One possible explanation for this disparity could be the associated costs and waiting times that families have to bear, despite immunizations being offered free of charge in Ethiopia. For families with limited financial resources, these travel costs and opportunity costs may be perceived as insurmountable obstacles, particularly in rural areas [26,30].

The status of household wealth emerged as a key determinant for zero-dose children in rural areas. Further analysis revealed that approximately 38% of the change in vaccination rates could be attributed to discrepancies in household wealth composition. Moreover, the analysis of CIs indicated [21] that the distribution of health services pertaining to zero-dose children favored the affluent, with a CI of −0.2250. This implied that the highest concentration of zero-dose children was observed among the poorest and less privileged households in the study locations. These findings align with previous studies conducted in Zimbabwe [31], Bangladesh [32], Nigeria [33], and India [34], which have consistently highlighted the association between household wealth and child vaccination status. One potential explanation for these wealth-related disparities in immunization coverage is that families with lower wealth may be compelled to prioritize income-generating activities to meet their basic needs and improve their standards of living [32]. Consequently, the pursuit of income becomes their primary focus, often at the expense of accessing immunization services. Another factor that might contribute to these disparities could be the health-seeking attitudes and practices prevalent among impoverished households, coupled with a lack of knowledge regarding the importance of vaccinations [32]. Additionally, individuals residing farther away from immunization centers may face logistical challenges, exacerbating the disparity in vaccination rates [35,36]. It is important to note that poverty and marginalization are widely recognized as major contributors to health inequalities, a stance supported by various scholarly works [36]. However, it is worth mentioning that the current health policy in Ethiopia lacks a comprehensive mechanism to compensate poor households for the time and transportation costs associated with accessing child healthcare services. This lack of a compensatory framework undermines the principle of equitable and inclusive healthcare provision for all target groups, especially those from disadvantaged backgrounds.

The wide disparity in coverage of zero-dose children in remote and underserved areas of Ethiopia can be attributed to various factors, as indicated by our study. Notably, the utilization of skilled birth attendants and institutional delivery at health facilities, along with ANC and family planning services, play a significant role in positively influencing the coverage of zero-dose children. Previous research also supports the idea that ANC checkups offer an opportunity to promote healthcare utilization among pregnant women. These checkups not only encourage institutional delivery but also emphasize the importance of vaccination and family planning. This is documented by other studies conducted in similar settings, which have found a strong association between ANC and increased coverage of zero-dose children [36,37,38]. There are several potential reasons for this association. Firstly, the interaction between pregnant women and community health workers can facilitate the uptake of vaccination and other healthcare services. Likewise, institutional delivery provides an opportunity to administer vaccinations immediately after birth, ensuring the child receives essential immunizations from the start. Furthermore, women’s engagement with healthcare providers and the health system during delivery and the post-partum period can enhance awareness of various health issues that may arise during early childhood [37,39,40,41]. 

Our findings also indicate that the educational background of mothers plays a significant role in the disparities observed in the vaccination coverage of zero-dose children. These disparities can be attributed to differences among socioeconomic groups in terms of where they reside. This is in line with previous studies conducted in India [39], which argue that higher levels of education among mothers leads to an increased use of healthcare services. Consequently, this facilitates the uptake of vaccines, ultimately reducing the number of children who have not received any doses in the areas studied. Our findings suggest that the unequal distribution of zero-dose children is influenced by three main factors: the level of education of mothers, the place where they reside, and their socioeconomic status. These findings are consistent with earlier research that demonstrated a correlation between higher socioeconomic status and a lower proportion of zero-dose children [40,41].

The results of our investigation showed an interesting relationship between maternal age and education and the prevalence of zero-dose children. Additionally, the result demonstrated that existing discrepancies in vaccine coverage were based on economic class. These disparities may be attributable to the accumulation of knowledge and experience over time. Our findings are consistent with previous research conducted in this field, adding weight and credibility to our study [38,42,43]. Decomposition analyses further supported our conclusion, revealing a positive correlation between child vaccination and maternal education within urban residences. This observation aligns with empirical literature advocating for improved maternal education to bolster child health outcomes worldwide [44,45]. One possible explanation for this phenomenon is that each additional year of formal education completed by a mother exerts a positive influence on the likelihood of her child receiving the essential vaccinations they need for protection against disease and infection. 

### Strengths and Limitations

In a groundbreaking analysis, our team delved into the vast database of the zero-dose project, uncovering invaluable insights into the socioeconomic disparities surrounding the immunization coverage of zero-dose children. This research not only offers a comprehensive overview of the current state of zero-dose children in Ethiopia, but also highlights the key factors driving these disparities at a population level.

Taking our investigation a step further, we ventured into underserved areas of Ethiopia to shed light on the previously unexplored inequalities in zero-dose children. This pioneering study meticulously analyzed the factors contributing to these disparities, providing a much-needed understanding of the complex dynamics at play in these marginalized communities.

However, it is crucial to acknowledge the limitations of our research. While our findings present a significant advancement in this field, it is essential to exercise caution when interpreting the results. Our study only examined a sample of the zero-dose population, and therefore, it would be inappropriate to generalize our conclusions to the entire child population in Ethiopia.

Additionally, given the nature of our data being cross-sectional, we want to emphasize that our study does not establish causality. Instead, it serves as a starting point for future research and as a call to action for policymakers and healthcare providers to address the underlying factors perpetuating the inequalities observed in zero-dose children.

With these findings, we aim to foster a greater understanding of the socioeconomic barriers hindering immunization coverage in Ethiopia. By illuminating the intricate web of factors contributing to these disparities, we hope to inspire evidence-based strategies that will ensure no child is left behind in the fight against vaccine-preventable diseases.

## 5. Conclusions

The study’s findings advocate for urgent action to address the alarming disparities in vaccination coverage observed among zero-dose children in remote, hard-to-reach, and underserved regions of Ethiopia. To substantially increase vaccination rates and decrease the number of zero-dose and under-immunized children in Ethiopia, it is important to implement context-specific and tailored strategies that directly address the socioeconomic barriers hindering vaccine accessibility. Key steps include the development and deployment of mobile health clinics to reach remote and underserved communities, ensuring vaccines are available at the point of use to alleviate vaccine shortage. Additionally, building robust partnerships with local community leaders and healthcare workers can foster trust and enhance community engagement, encouraging greater vaccine acceptance. Innovative approaches such as utilizing digital tools for health education and vaccination tracking can further support these efforts. By integrating these targeted strategies, Ethiopia can move towards more equitable healthcare outcomes for all children, regardless of their socioeconomic status.

## Figures and Tables

**Table 1 ijerph-21-01086-t001:** Total sample size and EAs required for the vaccination coverage survey in remote, hard-to-reach, and underserved settings of Ethiopia from 1 February to 31 July 2022.

Types of Study Population	Number of EAs	Sample Size
Afar	30	360
Somali	30	360
Benishangul Gumuz	30	360
Gambella	30	360
Newly formed regions Sidama and Southwest	30	360
Urban slums	40	480
Pastoralist areas in Oromia, SNNP, and Southwest	30	360
Hard-to-reach areas	30	360
Conflict affected areas	30	360
Refugees	30	360
Internally displaced persons	30	360
**Total**	**340**	**4080**

**Table 2 ijerph-21-01086-t002:** Sociodemographic characteristics of respondents in remote, hard-to-reach, and underserved settings of Ethiopia from 1 February to 31 July 2022.

Characteristics	Frequency (%)
Child’s sex	
Male	1985 (54.4)
Female	1661 (45.6)
Child’s age in months	
12–23	1849 (50.7)
24–35	1797 (49.3)
Age of mother/caregiver in years	
15–24	875 (24.0)
25–34	1969 (54.0)
35–44	572 (15.7)
≥45	105 (2.9)
Do not know	126 (3.5)
Educational status of mother/caregiver	
No formal education or preschool	2158 (59.2)
Primary education	788 (21.6)
Secondary education	616 (16.9)
Tertiary education	84 (2.3)
Marital status	
Not ever married	43 (1.2)
Married/Living together	3312 (90.8)
Separated	83 (2.3)
Divorced	110 (3.0)
Widowed	98 (2.7)
Place of residence	
Urban	677 (18.6)
Rural	2969 (81.4)
Employment status of mother/caregiver	
Unemployed	2098 (57.6)
Employed	1548 (42.4)
Region *	
Afar	636 (17.4)
Amhara	372 (10.2)
Oromia	431 (11.8)
Somali	480 (13.2)
Benishangul Gumuz	216 (5.9)
SNNP	300 (8.2)
Sidama	239 (6.6)
Southwest Ethiopia	181 (5.0)
Gambella	479 (13.1)
Harari	60 (1.6)
Addis Ababa	192 (5.3)
Dire Dawa	60 (1.6)
Household size	
2–5	2044 (56.1)
≥6	1602 (43.9)
Wealth index	
Richest	729 (19.99)
Richer	731 (20.05)
Middle	728 (19.97)
Poorer	729 (19.99)
Poorest	729 (19.99)

* Unweighted sample size.

**Table 3 ijerph-21-01086-t003:** CI results in remote, hard-to-reach, and underserved settings of Ethiopia from 1 February to 31 July 2022.

Index	No. of Obs.	Index Value	Standard Error	*p*-Value
Erreygers norm. CI	3646	−0.2250	0.01729332	<0.001

**Table 4 ijerph-21-01086-t004:** CIs, marginal effects, and contributions of covariates to inequality in immunization in remote, hard-to-reach, and underserved settings of Ethiopia from 1 February to 31 July 2022 (N = 3646).

Variable	Elasticity	CI	% Contribution
Wealth index			
Poorest	Ref	Ref	Ref
Poorer	−0.0790175	−0.0248157	−11.02817
Middle	−0.0899104	0.0077767	3.558421
Richer	−0.1232491	0.0516888	22.04815
Richest	−0.1125731	0.0548704	24.3477
Overall			38.926
Place of residence			
Urban	Ref	Ref	Ref
Rural	0.1337818	−0.0187361	7.690324
Child’s age in months			
12–23	Ref	Ref	Ref
24–35	0.0184178	0.000758	−0.4484304
Marital status			
Not married	Ref	Ref	Ref
Married/Living together	0.1023875	0.0015589	2.079752
Child’s sex			
Male	Ref	Ref	Ref
Female	0.0073494	−0.0000692	0.0053105
Household head’s sex			
Female	Ref	Ref	Ref
Male	0.2410311	0.0073635	−1.718179
Age of mother/caregiver in years			
15–24	Ref	Ref	Ref
25–34	−0.0369372	0.0010394	−0.6262123
35–44	0.0204524	0.0006407	−0.1870499
45 or above	0.0037871	−1.96 × 10^−6^	0.0013107
Caregiver’s employment status			
Not working	Ref	Ref	Ref
Working	0.055089	0.0025283	−1.764959
≥4 ANC visits during pregnancy			
Yes	Ref	Ref	Ref
No	−0.1157852	−0.019626	8.662506
PNC services for index child			
Yes	Ref	Ref	Ref
No	−0.2482841	−0.0183511	8.618182
Skilled birth attendance			
Yes	Ref	Ref	Ref
No	−0.1612879	−0.0435716	20.35813
Number of under-five children			
One child	Ref	Ref	Ref
Two children	−0.0531308	−0.0187361	−2.203588
Three and above	0.0259274	0.000758	0.6659956
Availability of health facility in the kebele			
Yes	Ref	Ref	Ref
No	−0.8824569	−0.1367844	61.02996
One-way walking distance to the nearest health facility			
Time to walk ≤ 30 min	Ref	Ref	Ref
Time to walk > 30 min	0.0173218	−0.0022571	0.4631407
Maternal education			
No formal education or preschool	Ref	Ref	Ref
Primary education	0.0306062	0.0011367	−0.3098591
Secondary education	0.0517623	0.011027	−4.494141
Tertiary education	−0.0001716	−0.0000108	0.0904441

## Data Availability

In the pursuit of advancing knowledge, the accessibility of data and research materials stands as a fundamental component. We understand the importance of open access and transparency in promoting cooperation and pushing the boundaries of scientific discovery. Consequently, we are committed to supporting scientific advancement by providing access to the data underlying this research to all interested entities upon reasonable request.

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
