# Peer review of "Understanding Socioeconomic Inequalities in Zero-Dose Children for Vaccination in Underserved Settings of Ethiopia: Decomposition Analysis Approach"

_ijerph, 2024, doi:10.3390/ijerph21081086_

Round 1

Reviewer 1 Report

Comments and Suggestions for Authors

The study  explored the association between zero-dose children coverage and various socioeconomic factors and quantified their contributions to generating inequalities in zero-dose children in remote, hard-to-reach, and special settings of Ethiopia. The comments are as follows:

1. The calculation/estimation of sample size should be presented. 

2. All the tables should be added in the main text of the manuscript. Besides, the authors should use three-line format to revise the tables. 

3. The reference style need to be revised one by one according to the format of the journal. 

4.  The study setting (line 87-113) can be shortened and the detailed information can be set as supplementary file. 

5. Dependent variable and independent variables (line 162-216) can also be shortened to make the words more concise. 

Comments on the Quality of English Language

The language and written need extensive revision.

Reviewer 2 Report

Comments and Suggestions for Authors

There are many studies done in various countries which is highlighted in the report. This research is country specific which ECHOs other developing countries findings. It is would be nice if authors also could outline steps that could be taken to increase vaccination in Ethiopia 

Reviewer 3 Report

Comments and Suggestions for Authors

Dear Authors,

Your article is very interesting and reveals important facts. Your conclusions will contribute to better vaccination coverage. However I have several comments:

Methods: you mentioned two times that the software is STATA. Please delete one of them, such a repetition is not necessary.

I would suggest to use other abbreviation for concentration index except of CI. The usual meaning of CI is confidence interval and even you used it within this article. It is confusing. Furthermore, the values of the index could be presented with up to 4 decimals instead of 8 decimals. For example the concentration index of -.22502133 that is reported on page 6 could be presented as -0.2250 or even -0.23

Results: The title of table 1 is confusing. The text clearly states that it reports information only about the zero-dosed children but its title suggests that is about the entire sample. I would suggest to report both zero-dosed and the other children in this table. Furthermore, the age of the respondents is not important, the age of the mother is of great importance. You could also test the mother’s age as a factor variable. The same is valid for the respondent’s education: mother’s education is important. Since >80% of the participants belong to rural type of residence you could explore its structure and try to divide rural to useful subtypes. Under the table is symbol explaining that the results are unweighted: please add an explanation about the weighting procedure you applied and a short explanation why some variables are not weighted.

In Table 3 you included some variables that should be reported also in Table 1: household’s head sex, N of antenatal visits, postnatal care services, skilled birth attendance etc.

I would suggest all the numerical variables (ages, numbers etc.) to be included as numerical variables into the analyses instead of categorizing them. Otherwise please give an appropriate explanation how did you select the cut-offs. I also would suggest the category “no formal education” not to be reference since it is already known that lacking of mother’s education is a factor for worsening of child’s health. The reference category could be tertiary education for instance.

You applied ordinary least square regression and reported the results in Table 3. Please add this method within Materials and methods section.
